# Sustainable HRM Practices in Corporate Reporting

Štěpánka Hronová * and Miroslav Špaček

Department of Entrepreneurship, Faculty of Business Administration, Prague University of Economics and Business, Nám. W. Churchilla 1938/4, 130 67 Praha, Czech Republic; miroslav.spacek@vse.cz
* Correspondence: stepanka.hronova@vse.cz

**Abstract:** Shaped by the current turbulent era of macroeconomic forces, inclusive of the technological challenges of Industry 4.0, and ubiquitous uncertainties, the business environment and its stakeholders hold high expectations for sustainable organizational practices, including harmonized and comprehensible sustainability reporting. Increasingly, responsible behavior towards internal stakeholders comes from within organizations, valuing employees as a key asset and introducing sustainable human resource management (S-HRM) practices to motivate their workforce. Reporting on these S-HRM practices and sustainability is in the highest interests of managers and investors alike. Focusing on the involved parties, employees most particularly, the paper contributes to the stakeholder theory. The literature review, previous S-HRM studies' interpretation and their critical assessment, the GRI standards' comparative analysis, and Lawshe's content validity approach have been applied as the methodological framework. With the purpose to extend the scientific literature on S-HRM and its reporting, the authors aim to close the gap between purely theoretical S-HRM treaties and more practically oriented studies on reporting. The findings on the key areas of S-HRM practices give rise to the S-HRM Practices Model, the main goal of this study. This comprehensible model may serve as a harmonized instrument for sustainable HRM reporting analysis and auditing for academia and practitioners alike.

**Keywords:** sustainable human resource management (S-HRM); non-financial reporting; GRI standards; sustainability; CSR; Industry 4.0; stakeholder theory; sustainable HRM practices model

## 1. Introduction

For many organizations worldwide, corporate sustainability has become a major focus because of changes in climate, regulatory pressures and overall demands for higher responsibility among companies regarding their impact on society and the environment (Cohen et al. 2012). This is not an easy task to achieve in these challenging turbulent times caused by uncertainties, continuous technological advancement, the increasing speed of innovation and the overall socio-economic, legal and political forces (Randev and Jha 2019). External pressures can influence the adoption of sustainable practices (Adebanjo et al. 2016). Increasingly, non-financial indicators come under the scrutiny of investors who are trying to identify the best investment opportunities. Other stakeholders, namely customers and the general public, show an increased interest in the sustainable behavior of companies. Through the application of sustainability principles at the employee level and their elaborated public disclosure, brands stand out from their competitors as responsible employers in the labor market. Consequently, the employer brand can be boosted.

However, it is not only the external pressures that have resulted in sustainability and responsible behavior of companies being incorporated into their daily operations as well as strategic documents. Exponentially, the interest comes from within organizations. In terms of sustainability, the firms are increasingly focusing on a comprehensive and well-integrated set of activities built into their business strategies and business plans (Porter and Krammer 2011), which may help internal stakeholders, the local communities or society as a whole. These proactive sustainable and responsible practices and business

strategies adopted by companies go beyond legal requirements and contribute positively to a satisfied society (Torugsa et al. 2012).

The sustainability concept has penetrated all management functions including strategic management, organizational behavior, supply chains and human resources (HR), and has gained attention from both the practitioners and academia (Randev and Jha 2019). It is mostly the latter, the HR function, that is critical to achieving success in implementing sustainability within organizations.

Sustainable management of employees, which are pivotal organizational assets, utilizes the HR tools and helps to embed the sustainability strategy into organizations (Cohen et al. 2012). The area of human resource management naturally reflects developments in society as well as trends in academic disciplines (Paauwe et al. 2013; De Prins et al. 2014). The goal of HR management is to provide motivated, qualified and loyal employees; in addition, sustainable human resource management (S-HMR) is the next logical step after incorporating sustainability into a company's processes.

> *"In this context, the key roles of Sustainable HRM are both to contribute to developing sustainable business organisations economically, ecologically and socially and to make HRM systems per se become more sustainable (Cohen et al. 2012; Ehnert and Harry 2012; Ehnert et al. 2014; Wilkinson et al. 2001)."* (Ehnert et al. 2016, p. 89)

Employees are crucial in relation to organizational development and success. Thus, it is of high importance that these internal stakeholders are treated ethically, fairly and responsibly.

Sustainable and ethical HR policies and their daily execution as well as their transparent disclosure is highly important to various stakeholders of a particular company influencing internal stakeholders, employees, and their overall well-being and satisfaction and the external actors alike as the transparent S-HRM is positively reflected in brand reputation and image. The topic of this paper, mostly concerning human resources and sustainable HRM practices, is closely connected with and contributes to the stakeholder theory further mentioned in the theoretical section.

Non-financial reporting of sustainability is of utmost importance to investors who can easily distinguish among the best investment opportunities. There are external and internal motivators for organizations to engage in sustainable behavior and new opportunities as well as trends in this dynamic field emerging in terms of evolving reporting guidelines and appearance of new communication media.

This paper covers the research avenue of S-HRM reporting, which has been sparsely examined so far. It is unique in its holistic approach to S-HRM and its disclosure in terms of novelty of information and the model it brings. The latter offers a harmonized compact tool for designating S-HRM practices, which has been called for by academia and practitioners alike. It can be successfully used in auditing or researching S-HRM practices published in corporate sustainability reports, just as it can increase their more comprehensive compilation.

In the next section of the *Introduction*, a broader theoretical view and a thorough literature review on S-HRM are presented first, followed by the state-of-the art trends in sustainability and HR reporting inclusive of its communication channels, resulting in the explanation of this paper's purpose, aim, goal and revealing the research question. Next, the *Methodology* section introduces the research scheme and describes the methods used. The paper continues with the *Results* section, showing the research process and its main findings, the *Discussion* section, interpreting and describing the significance of the findings, and it culminates in the *Conclusions*.

### 1.1. Literature Review

1.1.1. Anchoring the S-HRM Research on Theories

In a broader sense, anchoring the research of sustainable human resource management (S-HRM) in theories is designed eclectically, enabling us to compile a compact background framework. The topic is related to several theoretical viewpoints detailed below, but most importantly the paper contributes to the stakeholder theory, which best supports

the empirical part related to human resources and the sustainable HR management. The literature review is of a semi-systematic character (Snyder 2019), further described in the *Methodology*.

Firstly, starting with the broadest theoretical concept, in the theory of political economy, an organization exists as part of a larger social system and can be analyzed mainly in terms of macroeconomic factors which act as external forces that individual companies do not have the opportunity to influence. By considering this theory, a researcher is able to take into account the impact of broader societal issues on organizational operations as well as the selection of information intended for disclosure (Deegan 2011). From the viewpoint of sustainability, these issues include, for example, institutional support for this concept or statutory reporting. In a narrower sense of focus, it is possible to analyze the microenvironment of organizations: employees, competitors, community, suppliers, the media, and others.

Secondly, in its basic concept, this study draws upon the theory of organization, which perceives a company as an organized structure with interpersonal relationships, a given hierarchy as well as adaptive systems which have to adjust to various changes in their environments.

> *"Organization Theory deals primarily with the organization level phenomenon such as organizational change and growth, planning and design, development, politics, culture and structure."* (Haque and Rehman 2014)

Historically, organizational theories have developed from traditional to more modern concepts. Posey pointed out a discussion of a modern organization theory, attempting to explain

> *" . . . how interactions, activities, and sentiments within an organization are influenced by environment, which is classified into technical and physical, social and cultural, legal, and economic aspects."* (Posey 1961, p. 610)

The attention and focus of the neoclassical approach have shifted more to particular stakeholders such as individuals and work groups.

Thirdly, this study builds upon and mainly contributes to the stakeholder theory firstly published by Freeman in 1984 and reprinted in 2010 (Freeman 2010), which comprises morals and values reflected in business ethics and organizational management impacting various entities. The proper focus helps to create value for sustainability (Freudenreich et al. 2020). This theory explains the company's activities in meeting the needs of two key stakeholder groups: primary (owners, investors, employees, customers, suppliers, local authorities and the general public) and secondary (government, competitors, civic and trade associations, activists, etc.). Furthermore, the normative theory of stakeholder identification defines the specific stakeholders, while the descriptive theory of stakeholder salience looks at the particular conditions under which these parties are treated as stakeholders (Phillips 2003; Sustainable Human Resource Management with Salience of Stakeholders: A Top Management Perspective n.d.).

Another theory called resource-based view (RBV)/resource-advantage helps to define the strategic resources that a company has at its disposal and through which it can achieve a sustainable competitive advantage. Human capital is intended as the key and strategic resource helping to maintain the company's competitiveness (Wernerfelt 1984; Barney 1991).

According to legitimacy theory, the company is perceived in accordance with laws and social norms, and its activities ensure a certain legitimacy level of its operations (Suchman 1995). Non-financial reporting, including sustainability reporting with regard to HR, can be conceived here as an effort to justify the right to do business by emphasizing the contribution of the company to the community and society. Sustainability and S-HRM reporting can also be pragmatically perceived as a tool used to enhance the positive image and reputation of a company.

Next, the theory of signaling approach (Akerlof 1970; Spence 1973) looks at an entity transmitting information about itself to another entity as it tries to even the imbalance of information and help explain some uncertainty by signaling/revealing certain information. The authors Akerlof and Spence won the Nobel Prize for this concept in 2001. It is a significant elaboration and addition to the microeconomic theory. In his study, Starke et al. (2012) also drew attention to the asymmetry of information, which can be caused by inaccurate, tacit or scattered information. Entities on one side of the market have much better information than operators on the other; individual groups therefore differ significantly in the quality of information they possess and operate with. Reporting, including non-financial reporting on the sustainability of HR, can help to offset the imbalance with respect to the lack of publicly available information about the company's activities and procedures for the interested external stakeholders.

Finally, attention should be also paid to enablers of S-HRM, which are corporate ethics/ethical management and corporate governance (Hussain et al. 2018). The management strategy of the organization and its social environment are important factors for systems in which employees are respected and considered a key part of the company. Management that values their human resources can be called ethical (Fehr et al. 2015). Blanchard and Peale (1988) defined ethical management through the so-called 5Ps: the purpose or mission of the organization and reasonable pride, while the other factors are patience, perseverance and perspective, reflecting an understanding of what is important. The behavior of employees of the organization and corporate communication in external and internal space are dependent on the system of management and control processes in the organization. Corporate governance (especially for corporations and joint stock-companies) defines the obligations and rights between all stakeholders (shareholders, management, statutory bodies, HR, customers and possibly other interest groups).

The following section of the paper introduces the birth and development of the S-HRM concept in the literature.

### 1.1.2. The Journey from Strategic to Sustainable HRM

Strategic human resource management (S-HRM) involves the development of HR programs, practices and policies which are lined up with business strategy in order to achieve strategic objectives of a company.

The early definitions of human resource management in the literature frequently mentioned work practices ensuring high-performance, which is an attribute of strategic HRM. According to Kramar (2014), the concept of strategic human resources management (SHRM) was developed as a way of managing employees in the late 1970s and 1980s and arose in response to radical changes in an external turbulent environment; another stimulus was the ever-increasing size of companies. Armstrong et al. (2015) stated that strategic human resource management activities support the achievement of organizational goals and values by aligning human resource strategies with organizational strategy. Some authors, such as Legge (2005), criticized the so-called hard practices of HRM during the 1990s and supported the soft perspective of the HRM model.

Boselie (2014) sees the concept of sustainable HRM as a complement to the main principles of strategic HRM used since the 1980s, while De Lange and Koppens (2007) in De Prins et al. (2014) distinguished sustainable HRM from the mainstream HRM. De Prins et al. (2014) further presented a model based on three characteristics that can be assigned to Elkington's (1994) three Ps of sustainability (people, planet, profit) and used the abbreviation ROC, consisting of the words R = respect, O = openness and C = continuity. According to these authors, the sustainable HRM concept has the following characteristics: respect—restoring respect and consideration for the internal stakeholders in the organization, i.e., employees; openness—environmental awareness and HRM perspective from the outside; and continuity—a long-term approach to economic and social aspects and with regard to the employability of the individual. Mariappanadar thoroughly analyzed the newly emerging S-HRM definitions as follows:

> *"Early sustainable HRM definitions in the literature indicate the unsustainable impacts of high-performance work practices on employees such as reduced HR conservation (Mariappanadar 2003; Kramar 2014), reduced HR regeneration (Ehnert 2009b) and increased harm of work (Mariappanadar 2012, 2014b). The harm of work from the sustainable HRM perspective is about the restrictions imposed on employees by high-performance work practices that are designed to achieve organization financial performance. Furthermore, it was indicated that the harm of work is lined to obscured, reduced or lost psychological, social and work-related health well-being outcome for employees as internal stakeholders (Mariappanadar 2014a)."* (Mariappanadar 2020, p. 11)

There have also been ongoing debates among academics on online communication platforms on how exactly to define the attributes of S-HRM. ResearchGate is considered an up-to-date, real-time, fast and flexible communication tool and an accepted online scientific platform where academics and/or renowned scientific colleagues can meet to discuss issues, communicate concerns, share knowledge, collaborate, seek out answers, come to solutions and can obtain meaningful and valid results. The content is co-created by the users themselves, and thus reflects their real needs. For example, on this platform, there has been a discussion among researchers on what factors/practices really represent sustainable HR or how it correlates with S-HRM (ResearchGate 2021). Here are a few examples of academics discussing their concerns about S-HRM:

(a)  *"This requires conducting a thorough analysis on what factors or variables will determine whether HRM is sustainable. . . . However, this requires conducting an analysis of the HRM systems and/or HRM functions to measure, i.e., HR policy, practices, HR functions/activities to be measured for sustainability."* (A.A., Capella University, Minneapolis, MN, USA)

(b)  *"It really depends on how you define and operationalize sustainable HRM. However, if you mean things like pay level, heavy focus on training and development, highly selective recruitment, job security, and long-term HR-planning."* (B.K., BI Norwegian Business School, Oslo, Norway).

(c)  *"First to differentiate the two concepts of Strategic HRM and Sustainable HRM. Strategic HRM is more outward-looking and long-term decisions regarding HRM with the objective of winning competitive advantage over our competitors. While the fundamental objective of the sustainable HRM thinking is to develop employee loyalty and retention, the right HRM eco-system, and creating Quality of Work Life."* (N.A.A., Universiti Putra Malaysia, Putrajaya, Malaysia)

### 1.1.3. Sustainable HRM

A prerequisite for the incorporation of sustainable HRM practices is the overall application of the corporate sustainability (CS) concept, which is based on the wider framework of sustainable development (SD) defined in 1987 in a report called *Our Common Future* prepared by the World Commission of Environment and Development known as the Brundtland Commission (World Commission on Environment and Development 1987). The base for S-HRM, the SD, can be seen as the process described by Mazur and Walczyna (2020) as organization-driven change where the equally distributed attention to all three pillars of sustainability: the economic, social, and environmental are incorporated into organizational strategy. The concept of incorporating HR into sustainability and vice versa (as it is a reciprocal process) has evolved over the years and thus has changed the scope of the literature on the S-HRM as well as the understanding and definitions' shaping.

The first papers started to appear in the literature around the turn of the century and many of the below-mentioned academics have been actively publishing in the field of S-HRM up to now. In 2000, Gollan was among the first to use the term HR sustainability, Daily and Huang (2001) highlighted the role of HR in organizational sustainability, Forslin et al. (2002) utilized the phrase sustainable work systems, and Mariappanadar (2003) focused on sustainable human resource strategy in his paper where he defined the sustainable HR strategy as follows:

*"Sustainable HR strategy can be defined as the management of human resources to meet the optimal needs of the company and community of the present without compromising the ability to meet the needs of the future."* (p. 910)

One of the early explications of the sustainable management of human resources was provided by Ehnert (2009a), who indicated S-HRM as the next logical step coming after strategic HRM. She also introduced one of the most widely cited definitions where she sees the S-HRM:

*"As the adoption of HRM strategies and practices that enable the achievement of financial, social and ecological goals, with an impact inside and outside of the organization and over a long-term time horizon while controlling for unintended side effects and negative feedback."* (p. 90)

Kramar (2014) extended Ehnert's (2009a) above-quoted definition of S-HRM to practices that minimize the negative impact on the environment and the community as well. The author also emphasized the important role of CEOs, management, HR professionals, and employees in applying these practices. Sustainability in HR management according to App and Büttgen (2016) highlights the value of human resources and emphasizes the importance of employability and long-term availability of employees. It is a means of providing a qualified workforce, both for the current activities of the organization and future activities. The positive feedback of the application of a responsible and sustainable approach to HR brings a greater chance of success for the company thanks to these employees. Gollan (2005) correlates HR sustainability with high involvement management.

Nowadays, most authors agree that the sustainable management of companies and resources, including human resources, can help to adequately address the emerging situations in the macro-environment and micro-climate of organizations. Cohen et al. (2012) speaks of sustainable HRM as the use of HR tools to support the company's sustainability strategy and at the same time to create such an HRM system that will contribute to the company's sustainable performance. Stahl et al. (2020) supports the view of HR management having a vital role in contributing to corporate sustainability efforts.

Recently, academia has indicated a new research avenue based on Aust et al. (2020). According to their paper, a new level has been reached in scholarly debate about corporate sustainability, shifting focus from purely CS objectives to contributing to the sustainable development goals (SDGs) defined by the UN. It is a brand-new way of looking at the purpose of business organizations.

### 1.1.4. Stakeholder Oriented Sustainable HR with Focus on Employees

Organizational stakeholders are groups from the micro-environment who interact with the company and can have an influence on its operations or vice versa. Järlström et al. (2018) in their study on salience of stakeholders asked managers to define the stakeholders, which resulted in the following enumeration: owners, managers, customers, employees, labor unions and employee representatives. Other stakeholders can include suppliers, corporate partners, the local community, investors or control bodies. In their study, S-HRM was shown to have a connection to both internal as well as external stakeholders as follows:

*"The stakeholders identified in the justice and equality dimension were legislators, labor unions, and employer organizations. The stakeholders identified in the transparent HR practices dimension were managers who implement HR practices, and the employees who are the target of those practices. The stakeholders identified in the profitability dimension were owners, and also managers, because they are involved in creating business strategies and are responsible for the overall success of the organization. Finally, in the employee well-being dimension, the employees are the main stakeholders, but managers and supervisors also play an important role as actors who are safeguarding employee well-being."* (Järlström et al. 2018)

The increasing need for addressing social and HR sustainability issues and social performance drivers (Ioannou and Serafeim 2012) during the decade after the turn of the

century brought about more studies published on S-HRM with focus on employees, managers and internal communication. Studies focusing on building sustainable organizations emphasized the human factor. Furthermore, employee-centered studies appeared (Pasban and Nojedeh 2016) solving various aspects of S-HRM including HR policies, processes aimed at employees to implement various strategies and organizational changes (Eccles et al. 2014), implementation of sustainability practices on the employee level, diversity, equality, well-being of employees, education, optimizing employability (Veld et al. 2015), and others. Practices of HR management and HR development were argued to play an important role in responsible behavior (Jamali et al. 2015) and sustainability implementation (Ardichvili 2013). The below diagram (Figure 1) shows some key HR sustainability studies and their topics chronologically.

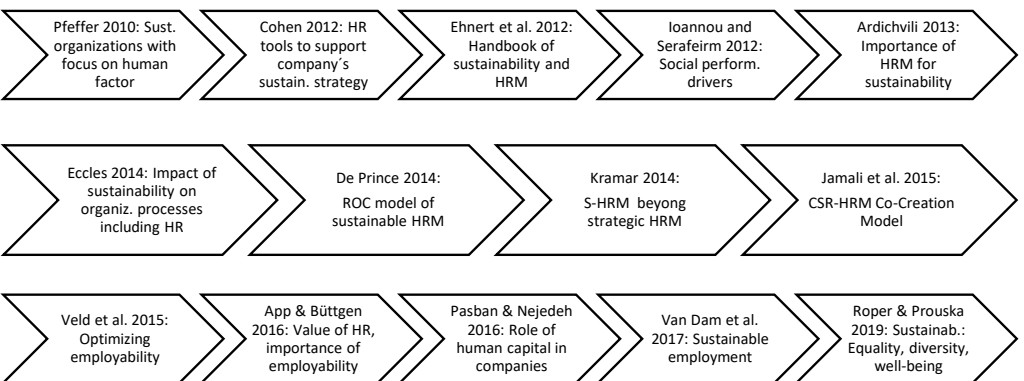

**Figure 1.** Topics of sustainability in HR during the second decade of the 21st century.

In 2018, Stankevičiūtė and Savanevičienė published a study identifying several areas most frequently mentioned in theoretical papers in connection with sustainable HRM. The areas they revealed included: long term-orientation; care of employees; care of environment; employee participation; profitability (shared); employee development; external partnership; flexibility; compliance beyond labor regulations; employee cooperation; and fairness and equality. This study served as one of the inspirational resources for the Sustainable Human Resource Management Analytical Framework introduced later in this study; however, it does not have a connection with the issue of reporting.

Several recent studies e.g., in Roper et al. (2020) have also highlighted organizational structure, processes and strategies as important factors impacting internal stakeholders, employees and influencing various S-HRM aspects. For example, Richards (2020) suggests the employee-centric rather than employer-driven approach to corporate sustainable practices. Lopez-Cabrales and Valle-Cabrera (2019) focus on the adoption of a sustainable corporate approach which involves developing coherent HRM strategies translated into sustainable HRM systems where these strategies require new employment relationships.

Progressively, an issue of HR development (HRD) is seen as an element of sustainable HRM or responsible corporate practices (Jang and Ardichvili 2020). Some scholars also extend the sustainable management of HRD beyond boundaries of organizations, looking at various types of employment tiers with the aim being to contribute to the community development (Li 2020), or they combine the sustainable HRD with the necessity to provide education and training to HR in connection with the concept of the Industry 4.0 (Piwowar-Sulej 2021).

Another research avenue focusing on internal stakeholders deals with the topic of employee representation (e.g., by union or an individual) for the purposes of employee-management negotiations about working conditions, wages, benefits or working hours. Debono (2017) in Prouska et al. (2019) sees organization structure as a key factor affecting employee attitude towards their representation. Prouska et al. (2019) enumerates elements with an influence on employee representation. Besides membership in labor unions,

the author names socio-economic variables such as education, occupation, gender and political beliefs.

Increasingly, HR practices and processes as well as their impact on employees are in the more detailed scope of interest. Mariappanadar (2020) sheds light on the difference between the terms S-HRM characteristics and S-HRM practices. Based on Posthuma et al. (2013), Mariappanadar (2020) describes HRM practices as specific methods which are adopted by companies in order to achieve business goals via implementation of the values and policies of an organization. Mariappanadar (2020) also demonstrates, with the use of Madden et al. (2012), that it is of high importance to evolve and grow proper characteristics of HRM practices so that the business strategies can be operationalized and the competitive advantage can be achieved. Therefore, the characteristics of the practices are "*about the underlying organizational motives perceived by employees who shape their behavior and attitude at work*" (Mariappanadar 2020, p. 12).

One of the newest publications on S-HRM is the *Sustainable Human Resource Management—* Transforming Organizations, Societies and Environment by Vanka et al. (2020) which evaluates the emerging policies and existing practices relating to S-HRM. The book also includes a contribution by S. Mariappanadar, one of the authors first publishing on S-HRM, who discusses characteristics and practices of S-HRM.

Aust et al. (2020) introduces the brand-new direction of research avenue in academia with focus on the Sustainable Development Goals (SDGs) proposed by the UN. The author claims that these goals challenge the purpose of business organizations and thus also the HRM.

### 1.1.5. S-HRM Reporting

External as well as internal pressures exist on sustainability reporting and increasingly it is in the vested interest of companies in disclosing information about their HR policies, strategies and overall treatment of their internal stakeholders, employees.

Reporting about S-HRM is voluntary unless the organization falls under the EU Directive 204/95/EU. There are numerous reporting standards (described in the next sub-chapter in detail) which can be followed in terms of non-financial disclosure. It is important to know what key characteristics or practices of the sustainable human resource management to look for when auditing corporate reports or when compiling its section with S-HRM in mind. There are very few studies concentrating on S-HRM reporting.

Some authors originally publishing purely on S-HRM turned their focus partly on S-HRM reporting such as Ehnert et al. (2016) or Jang and Ardichvili (2020). Here, a significant gap in the research has been identified, and the authors' aim is to fill it with their study, and thus inhabit this deserted field of research.

The following sub-chapter provides thorough information on various aspects of sustainability reporting inclusive of the newest trends and changes in this field of non-financial disclosure.

### 1.2. State-of-the-Art Sustainability and HR Reporting

Responding to a stronger awareness and sensitivity to the assisting role companies are increasingly playing worldwide in ecological, social and economic problems, a growing readiness exists for the biggest corporate players worldwide to show their commitment to corporate sustainability (Ehnert et al. 2016). Increasing numbers of organizations thus appear willing to disclose information on the three pillars of their performance in the economic, social and ecological areas (Schaltegger and Wagner 2006).

Additionally, due to the increasing external pressure on sustainability reporting (Hahn and Kühnen 2013), more companies gradually began to respond to the situation by publishing information on various areas of sustainability associated with the company's activities (KPMG 2011, 2013; Corporate-Responsiblity-Reporting-2012-Eng.Pdf n.d.). In general, organizations report on their activities for greater transparency and for building a wider awareness of their sustainable activities (Roberts 2009).

Most of the content of this paper has been written to provide general implications applicable worldwide; however, in some parts, it focuses on the Czech Republic—e.g., in terms of reporting tools and model situations which can be inspirational and enriching for companies in other countries. Based on the KPMG Sustainability reporting survey, sustainability reporting in the Czech Republic (CR) is sharply increasing and gradually gaining more importance. It has increased by 15 percent since the last analyzed period 3 years ago (Půrová and Dušek 2019; KPMG 2017). Currently, 66% of the largest companies in the CR report about their sustainable activities (KPMG 2020).

### 1.2.1. Types of Reports and Reporting Guidelines

Hetze and Winistörfer (2016) found that the naming of the document disclosing non-financial information about companies may differ with respect to geographical location. In Europe, the title Sustainability Report is preferred, while in the US and Asia, the term CSR Report is rather used. The authors of this paper have come across many other various titles being used for documents of this type for foreign-owned organizations in the Czech Republic. These are, for example: Impact Reports, Accountability Reports, Corporate Citizenship Reports, Integrated Reports or Triple Bottom Line (TBL) Reports. In the past, there used to be mainly individual reports focused either on environment or social matters which accompanied the commonly published annual reports. There has recently been a shift from stand-alone social or environmental reports with a sustainable focus to more complex comprehensive reports (Hahn and Kühnen 2013).

Reports on the sustainable and responsible behavior of companies cover many different aspects and relate primarily to ethical and transparent conduct that contributes to fair practices towards internal and external stakeholders, healthier and more prosperous societies and the improvement of the current state of the environment. In terms of disclosing information on sustainable HR, according to Forbes, one of the comparative studies among the 250 best companies in the world showed that in focusing on reporting HR activities, companies report data of internal employees rather than employees in the supply chain (Ehnert et al. 2016).

Non-financial reporting can be prepared according to the guidelines, standards and recommendations of world-renowned institutions. Among the best known across the globe are, for example, the United Nations (UN) and their UN Global Compact Initiative, Organisation for Economic Co-operation and Development (OECD), Global Reporting Initiative (GRI) and their Sustainability Reporting Standards or International Organization for Standardization (ISO) and their ISO 26000 or ISO 14000 norms. Other reference standards include SA 8000, issued by the non-governmental international organization Social Accountability International (SAI), and the EMAS methodology, developed by the European Commission as a voluntary tool for environmental management.

Based on the KPMG (2020) report, the GRI remains the most commonly used standard framework for reporting. It was used by around two thirds of N100 reporters and around three quarters of G250 reporters. A significant increase is reported in the application of the GRI Standards, which were introduced at the end of 2016.

In the Czech Republic in 2013, the Office for Technical Standardization, Metrology and State Testing issued the Czech standard ČSN 01 0391, which in addition to general topics of sustainable corporate behavior and CSR management also solves communication with external stakeholders. It includes the principles of company behavior, which are specified in international standards and documents of the following organizations: UN, OECD, ILO, EU. The standard can also be used for certification and defines the requirements for a social responsibility management system in the public and business sectors.

The communication focused on external stakeholders may lead to various benefits for companies and organizations (increased visibility and enhanced awareness may serve as examples). More exemplars are stated below.

1.2.2. Reporting Focused on Human Resources

In the Czech Republic, various guidelines are used by companies for reporting about factors related to the social aspects of sustainability. For example, according to the GRI standards used by some companies in the Czech Republic, HR in sections about labor law procedures and decent working conditions disclose information on, for example: *Employment of workers; Relationships between employees and management; Health and safety of employees; Training and education; Diversity and equal opportunities; Gender equality in pay; Assessment of suppliers in terms of work procedures; Complaint mechanisms for working procedures* (The Global Standards for Sustainability Reporting 2021).

The ISO 26000 standard, which provides methodological guidelines on the basic principles of social responsibility and sustainability, is relatively well-known in the Czech Republic thanks to its available version in the Czech language. In general, it aims to provide guidance on functioning CSR/sustainability in companies, identify and involve all stakeholders, increase the credibility of the organization, and comply with existing documents, agreements and codes, such as the International Labor Organization (ILO) or the World Declaration of Human Rights. In the HR sections, it focuses on corporate governance, human rights and labor relations. Further, human rights are divided into: due diligence; situations where human rights are at risk; avoidance of participation; complaints handling; discrimination and vulnerable groups; civil and political rights; economic, social and cultural rights; basic principles of law at work. In the sections of labor relations, the standard proposes to disclose information on the following: employment and labor relations; working conditions and social protection; social dialogue; safety and health protection during work (Moratis and Cochius 2011).

The previously mentioned Czech technical standard ČSN 01 0391, entitled *Corporate Social Responsibility Management System—Requirements*, can be used in all types and for all sizes of organizations. This standard, which can also be used for certification, defines the requirements for a social responsibility management system in the public and business sectors. In the section entitled Communication, it deals with external communication, i.e., the possibility of reporting on sustainable activities, including activities focused on the social aspect of sustainability and development of employees (ČSN 01 0391 2013).

1.2.3. Changes in Reporting

Over the last five years, major changes have occurred in the field of reporting. For example, in mid-2017, the AA1000 standard—AccountAbility Principles Standard from 2008 were updated to inform about corporate responsibility (AccountAbility n.d.). Another innovation is the transition concerning the Global Reporting Initiative reporting. There was a transfer from the G4 guidelines to the so-called GRI Standards, which fully replaced the previous standard in June 2018.

Since 2017, there has been a newly introduced reporting obligation in the EU for organizations that are public interest entities with more than 500 employees according to Directive 2014/95/EU of the European Parliament and of the Council of the European Union (Směrnice Evropského Parlamentu a Rady 2014/95/EU Ze Dne 22. . . .—EUR-Lex n.d.). This directive encourages companies to use some of the above reporting standards and lists the areas that the report should focus on. In the context of social and employment issues, these include: measures to ensure gender equality, implementation of core ILO conventions, working conditions, social dialogue, respect for workers' right to information and consultation, respect for trade union rights, health and safety at work, local communities, and measures to ensure the protection and development of those communities. With regard to human rights and the fight against corruption and bribery, the overview of non-financial information could include information on the prevention of human rights violations or on tools to fight corruption and bribery (Directive 2014/95/EU).

### 1.2.4. Reporting and Communication with Stakeholders Using the New Media

Firms seek to disseminate information and educate their potential or existing customers and other stakeholders about their sustainable activities. The focus of organizations is on a comprehensive set of activities incorporated into the business strategy and business plan (Porter and Krammer 2011), and there is a need for these to be communicated within the company as well as outwards. With the emergence of new media, organizations increasingly inform stakeholders about these activities in an electronic form on their websites or social networks, for example in the form of downloadable files and brochures, interactive reports, educational videos about the organization's activities, infographics or charts on the organization's website. Additionally, with the occurrence of Industry 4.0, there are new emerging technologies such as big data and artificial intelligence (AI) which companies inclusive of HR should be preparing for (Sivathanu and Pillai 2018).

Communication with stakeholders more frequently takes the electronic form of published reports. Due to the fact that many people increasingly obtain electronic information via mobile devices rather than from fixed desktop computers, companies also use so-called responsive web design and allow those interested to obtain this information via mobile phones or other electronic communication devices such as tablets, laptops, netbooks, etc., in a user-friendly format. Stakeholders draw on information publicly available in social media, which enables not only two-way communication, but also, above all, higher engagement. Freeman and Moutchnik (2013) state that social media accelerates the process of stakeholder involvement and enables the participation of a much wider range of stakeholders. Thanks to new technologies combined with the creativity of experts, we can see a number of examples, which are the proof of the fact that if corporate sustainability strategy is really thoughtfully implemented into the company's business strategy, it has a huge potential. For example, mobile applications are becoming more and more widespread worldwide, which can encourage responsible behavior with external and internal stakeholders at the initiative of the company. Here are some examples: *Apps For Good*, *SpillMap*, *Catalista*, *SeeClickFix*, *Eco Hero*, *GoodGuide*, *CauseWorld*, *Give Work*, and others (Woyke 2010). These interactive modern approaches to communication with stakeholders are not only the domain of the non-profit sector, but they are of a concern of the corporate sector as well. For example, the *EPP Help with Movement* mobile application from the ČEZ Foundation (a Czech company generating, distributing and selling electricity and heat) will serve as an example of a local innovative approach involving employees and the public in responsible activities, both in terms of human responsibility towards oneself—one's own movement and health care—and in terms of charity when the acquired points can be contributed by participants to a good cause (Bačuvčík et al. 2016).

### 1.2.5. Advantages of Sustainability and HR Reporting

Today, sustainability reporting is changing its nature. It used to have the form of ad hoc or rather exceptional non-financial reporting. However, information published in sustainability reports are of critical importance to investors who can easily distinguish among the best investment opportunities, recognizing the business managements with acumen. The authors of this paper are convinced that independent, unbiased and impartial guides in the form of a focused set of generally recognized standards would provide business managers with an easily applicable tool. On the path to sustainability, the Global Reporting Initiative (GRI), which deals with standardization of reporting, has helped companies in their reporting efforts; however, there are also many other initiatives focused on new goals.

State-of-the-art sustainability reporting not only helps capital providers; it is of utmost importance for all stakeholders of a particular company. Seeing the world through the lens of sustainability reveals the real needs that require attention, and this might be the source of innovations for companies. In his article for MIT Sloan Management Review, Andrew Winston noted:

*"Companies that innovate to solve environmental and social challenges create products and services that customers want and feel good about. None of what I'm talking about is philanthropic—it's all about business value."* (Winston 2018)

In the long run, these activities will be positively reflected in brand image and brand awareness. Additionally, investments into innovations create business value. Thus, financing sustainability activities should not be primarily seen as costs but rather as investments for the better future of companies, local communities and stakeholders.

The Global Reporting Initiative (GRI) divides the benefits of sustainable reporting into internal and external benefits. Among the internal advantages of sustainability reporting are: better understanding of risks and opportunities, emphasizing the link between financial and non-financial performance, influencing the long-term strategy and management policy of organizations and business plans, simplification of processes, costs reduction and improved efficiency, comparing and evaluating the sustainable behavior of a company with respect to laws, standards, codes, performance standards and voluntary initiatives. There is also the possibility to avoid negative publicity regarding failures in the environmental, social and managerial fields. Last but not least, comparison of organizational performance internally and between organizations and sectors helps companies in better planning their sustainable policies. Among external advantages, there are: mitigating or reversing the negative effects on the social environment and the governance of the organization; improving brand reputation and loyalty; enabling external stakeholders to understand the real values of the brand, tangible and intangible assets; demonstrating the impact of the sustainability concept on the company and at the same time introduce the company's sustainable behavior. These are avenues of possible engagements of companies in sustainable behavior, as seen today. However, new opportunities are ever emerging in this dynamic field.

1.2.6. Sustainability and Reporting Practices under Criticism

There are also critical voices against the concept of sustainability and its reporting. Often, they concentrate on the overload of information, the so called "green-washing" about environmentally friendly products or activities of the company which might not necessarily be true, and which may convey false impression that goods or processes are ecological. According to the authors, protection against such accusations seeks to support the claims with indisputable facts.

The critics of the concept of sustainable development mention above all the still high level of consumerism and indebtedness (states, companies and individuals), waste of raw materials, and constant burden on the environment. Rather, they encourage the economy of sharing, the circular economy, and they see non-growth as a starting point.

Others, such as Pfeffer (2010), challenge the disproportionate importance of the various components of sustainability, arguing that environmental sustainability is often the focus of excessive attention, while sustainable human resource management (working hours, redundancy policies, working conditions, etc.) is lagging behind. Calow (2013) draws attention to the imbalance between the three pillars of sustainability and questions the optimal rate of growth or reduction for one component without negatively impacting the other two pillars of sustainable development (e.g., the production process reduces to inputs of energy and resources, but this can cause imbalance of other components or have a negative impact on other parts of the company's operations). He offers the idea that market forces and the monetization of certain goods can help the sustainability process.

Benson and Craig (2014) criticize the concept of sustainability by claiming (especially with respect to ecology and natural resources) that humanity has already crossed the sustainability threshold and caused irreversible changes in biodiversity, global climate change and an exponential increase in per capita resource consumption. They claim that there is a phase of resilience, the need to adapt to the conditions that have already occurred. They base their claims on reports from the United Nations Environment Program (UNEP 2012).

Some authors consider (Crowther 2004) or criticize the so called "window-dressing" in reporting based on the GRI standards (Boiral 2013; Milne and Gray 2013). Journeault et al. (2021) also contribute to the debate regarding GRI standards, which they accuse of having negative aspects, overshadowing and silencing alternative ontologies.

While in accounting as the basis of financial reporting companies can use legally valid accounting principles for "window-dressing", doing something such as this is virtually impossible in sustainable non-financial reporting where companies report clearly and openly about actions which actually took place. The communication embodied in non-financial reports has to be transparent since in the time of social media, organizations can hardly hide anything without the risk that it would be disclosed sooner or later.

Having gone through the theoretical base, literature review and background information on sustainability and HR reporting, while also introducing the critical views on the issue, the final part of the *Introduction* reveals the purpose, aim, and goal of this study, inclusive of its added value. It also leads the readers to the paper's research question.

### 1.3. The Study's Added Value, Purpose, Aim and the Research Question

The added value of this paper lies mostly in its main findings on the S-HRM practices which enabled fulfillment of its goal: elaboration of a standardized and validated model for evaluating sustainability reports with respect to S-HRM.

The sustainability reporting is not legally enforceable (except for the Directive 2014/95/EU) and can come in various formats. Even though, according to KPMG (2020), about 77% of the largest companies in the world report on sustainability using some kind of reporting standard, it is clearly stated that they are not harmonized. Therefore, a goal of the paper was to provide a standardized model for reporting S-HRM practices which can help various stakeholders make their qualified decisions—e.g., potential employees or investors.

Most of the content has been written to provide general implications to be used worldwide; however, in some parts, it focuses on the Czech Republic—e.g., in terms of reporting tools and model situations which can be inspirational and enriching for companies in other countries. Sustainability reporting in the Czech Republic (CR) is sharply increasing and gradually gaining more importance based on the KPMG sustainability reporting 2020 survey. It has increased by 15 percent since the last analyzed period 3 years ago. Currently, 66% of the largest companies in the CR report about their sustainable activities (KPMG 2020).

The purpose of this paper is to broaden and update the literature on sustainable human resource management and its non-financial disclosure in sustainability reports and/or other corporate documents as it focuses on the S-HRM reflected in published studies, monograph and sustainability reporting guidelines and documents. The originality of the study lies in the proposed and below-described Sustainable HRM Practices Model.

Abductive analysis is applied, which enables forming conclusions from the information which is known: the literature review and state-of-the-art reporting guidelines. Based on literature review and qualitative comparative analysis of Global Reporting Initiative (GRI) guidelines, the gap was identified in the lack of studies on S-HRM reporting and particular S-HRM practices to be found in non-financial disclosure so that an audit of sustainable HRM practices of companies can be made. The aim is to close this gap. As various formats and types of delivery of sustainability reporting exist with the absence of legal enforceability of such disclosure (except for the Directive 2014/95/EU), the need for a harmonized comparative measure is urgent. Thus, the below research question is proposed which shall enable the main findings to be revealed and the goal to be fulfilled:

RQ: What are key areas for auditing sustainable HR practices in non-financial statements?

This paper aims to present the results in the form of an analytical framework and derive a validated tool from it. The goal is to elaborate a model of sustainable HR practices which will represent the main findings. A scheme of the research process is presented in the

*Methodology* section. With the defined practices of S-HRM, this model could be further used as a tool for auditing sustainable HRM practices based on disclosure published in corporate documents such as annual reports, impact reports, sustainability and CSR reports, etc. including communication via tools and platforms of the new media.

Academia can draw from the results and main findings of this study in terms of S-HRM attributes/practices definitions and the possibility to use the available validated tool, the Sustainable HRM Practices Model, for auditing corporate documents in terms of sustainable HRM practices as there is some uncertainty about the scope and delimitation of the S-HRM practices as shown in the *Introduction*.

In general, this study's aim is to help in bridging the gap between the partly overlapping areas of theoretical papers on S-HRM and the more practically oriented studies focused on sustainable reporting. This study does not deal with specific economic aspects of sustainable corporate behavior with focus on S-HRM.

## 2. Methodology

The purpose of this paper is to broaden and update the literature on sustainable human resource management and its aim is to close the gap on a specific research avenue: sustainable HRM reporting. The main findings will represent answers to the research question: *What are key areas for auditing sustainable HR practices in non-financial statements?* In this study, the applied type of analysis was abductive, enabling shaping conclusions from the information which is known: the literature review and state-of-the-art reporting guidelines.

Three main methods were used in this study to serve the defined purpose and goal, to achieve the set goal, to answer the research question and thus to arrive at the main findings. These were the semi-systematic literature review, the qualitative comparative analysis and the content validity test. These carefully chosen methodologies corresponded with the purpose and goal of the paper and the rationale behind the selected methods is described below. A visual representation of the methodological model is presented in Figure 2.

First of all, literature review was chosen as a methodological tool for the state-of-the-art knowledge mainly on the topic of Sustainable HRM (S-HRM).

> *"An effective and well-conducted review as a research method creates a firm foundation for advancing knowledge and facilitating theory development."* (Webster and Watson 2002 in Snyder 2019)

Literature review as a research tool is recommended especially for areas evolving at an increasing speed and is relevant for gathering data from business environment (Snyder 2019), which gives grounds for the relevance of this method used in this particular study focusing on corporate sustainability and S-HRM. Various authors view the literature review as a research method creating firm foundations for advancing knowledge and enabling theory development, seeing it as a more or less systematic way of gathering and synthesizing previous research (Baumeister and Leary 1997; Tranfield et al. 2003; Snyder 2019) The semi-systematic literature review is standing between two other models: the systematic and integrative model. Snyder (2019) further specifies the model (Table 1) building on the ideas of Wong et al. (2013):

> *"Besides the aim of overviewing a topic, a semi-systematic review often looks at how research within a selected field has progressed over time or how a topic has developed across research traditions. In general, the review seeks to identify and understand all potentially relevant research traditions that have implications for the studied topic and to synthesize these using meta-narratives instead of by measuring effect size."*

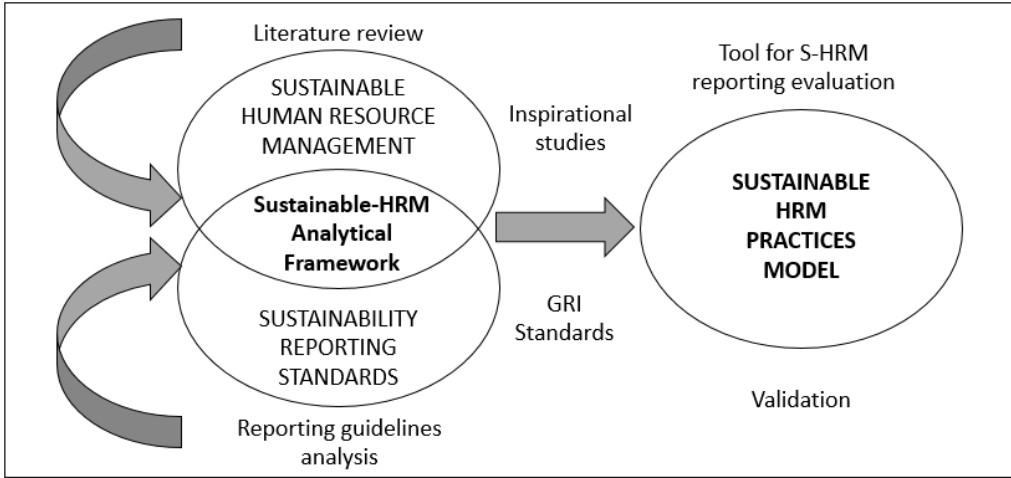

**Figure 2.** Methodological Model.

This methodological tool fitting the purpose and goal of this study uses scholarly sources on a specific topic and provides state-of-the-art knowledge in the field of social aspects of sustainability with a focus on sustainable human resource management. It allows us to identify relevant theories creating a wider framework, particular research avenues and gaps in the existing research with a focus on sustainable HRM (S-HRM) and sustainability/S-HRM reporting. The aim is to close the identified gap.

**Table 1.** Semi-systematic approach to literature review (Snyder 2019), shortened by the authors.

| Approach | Semi-Systematic |
|---|---|
| Typical purpose | Overview research area and track development over time |
| Research questions | Broad |
| Search strategy | May or may not be systematic |
| Sample characteristics | Research articles |
| Analysis and evaluation | Qualitative/quantitative |
| Examples of contribution | State of knowledge |
| | Themes in literature |
| | Historical overview |
| | Research agenda |
| | Theoretical model |

Second, qualitative comparative analysis of the Global Reporting Initiative guidelines content was used to look for specific subject areas when analyzing the standards. Bryman and Bell (2015) recommend this research tool for analyzing documents and texts which were either produced by companies (e.g., annual reports) or written about them (e.g., articles in the business press). This type of analysis is sometimes called ethnographic content analysis. Bryman and Bell (2015, p. 300) claim that

*"there is an emphasis on allowing categories to emerge out of data an on recognizing the significance for understanding meaning in the context in which an item being analyzed (and the categories derived from it) appeared."*

Literature review and the qualitative comparative analysis provide for a Sustainable HRM Practices Framework—identifying the complex S-HRM practices as well as showing possible keywords and offering a wider focus for further quantitative-qualitative content analysis of financial reports. Out of this Sustainable HRM Practices Framework, the S-HRM Practices Model is derived, representing the main findings of this study. The validity of the model was tested as described below.

Third, the content validity test was applied to examine the validity of the proposed S-HRM Practices Model generated based on the literature review and the GRI standards' analyses. This method is proposed in order to examine the validity of the tool (Cahyaningtyas et al. 2021).

It is a method for gauging agreement among expert evaluators regarding how essential each item/variable is. The larger the number of experts agreeing on the essentiality of the item of out 3 criteria (here: essential, non-essential and not applicable), the greater the level of content validity that exists (Lawshe 1975). When validating an item, the content validity ratio (CVR) is computed. Table 2 shows the given minimum levels for various numbers of panelists presented by Lawshe. Only the items which meet these minimum values should be used. Below, a reduced version of the table is presented. The original table by Lawshe includes odd numbers as well and starts at the minimum number of 5 panelists, thus the acceptable number of panelists is ≥5. In this study, the tool is used to validate the proposed S-HRM Practices Model.

**Table 2.** Minimum values of CVR and CVR$_t$. One-tailed test, *p* = 0.05 (Lawshe 1975).

| No. of Panelists | Min. Value | |
|:---:|:---:|:---:|
| 6 | 0.99 | [1] (1.00) |
| 8 | 0.75 | |
| 10 | 0.62 | |
| 12 | 0.56 | |

[1] When all panelists evaluate the criteria as essential, the CVR is computed to be 1.00 but for the ease of manipulations it is adjusted to 0.99. The smallest number of panelists in the model is 5.

In the content validity test, the content validity ratio is calculated for each of the proposed items based on the following formula

$$CVR = \frac{Ne - \frac{N}{2}}{\frac{N}{2}}$$

where *CVR* is the content validity ratio, *Ne* is the number of expert panelists giving assessment "essential/crucial" to the evaluated variable, and *N* is the total number of experts in the Content Evaluation Panel.

After the items have been evaluated and confirmed, the Content Validity Index (CVI) can be computed for the whole test. The CVI is the mean of the CVR values.

With quantification of the given criteria based on quantitative content analysis of the given reporting documents, pilot testing of the model will be run in further research. The S-HRM practices index will be calculated and the proposed index classification will be verified in terms of the response consistency. The results will be analyzed with the use of the Cronbach's alpha formula

$$\alpha = \frac{N \cdot \bar{c}}{\bar{v} + (N-1) \cdot \bar{c}}$$

used for calculations for reliability testing, where *N* is the number of items, $\bar{c}$ is the average covariance between item-pairs, and $\bar{v}$ is the average variance.

Based on the normal statistical distribution, it will be further assessed how the company is performing in terms of S-HRM practices compared to other analyzed subjects.

## 3. Results

The results of this study are presented in two sub-chapters. Section 3.1 presents the S-HRM Analytical Framework, while sub-chapter 3.2 introduces the S-HRM Practices Model, a validated instrument streamlining the Framework by defining eight key areas of S-HRM practices.

### 3.1. S-HRM Analytical Framework

Based on the GRI Standards, S-HRM literature review and SCOPUS indexed studies, a S-HRM Analytical Framework has been devised (Table 3).

The elaboration of Table 3 shows the gradual emergence of the further proposed S-HRM Practices Model, highlighting eight practices and proposing key words in context to concentrate on when executing a qualitative-quantitative analysis of sustainability reports with an intention to audit the corporate S-HRM practices. In particular, the Analytical Framework is based on a S-HRM literature review and a comparative analysis of the GRI 3, 4 and GRI Standards guidelines, a study presented by the Society for Human Resource Management called *HRM's Role in Corporate Social and Environmental Sustainability* (Cohen et al. 2012) and inspired mainly by two recent SCOPUS indexed studies. One of them focuses on sustainable HR development reporting (Jang and Ardichvili 2020). The other one is a theoretical paper on defining S-HRM characteristics (Stankevičiūtė and Savanevičienė 2018) but lacking the ties to non-financial reporting.

GRI in the chart shows GRI guideline areas corresponding to S-HRM, while the next two columns from the left indicate particular standards and possible keywords for the qualitative-quantitative analysis of non-financial reports, another column judges the compliance of the GRI standards areas with S-HRM characteristics defined in the theoretical study by the Stankevičiūtė and Savanevičienė (2018). The next column indicates the relevance of these characteristics for this study. The last column on the right-hand side of the table proposes other indicators/keywords for further qualitative-quantitative analyses of S-HRM disclosure. Horizontal lines with abbreviation S-HRM red and numbers 1–8 detail the eight practices used in the process of creating the S-HRM Practices Model, presented in Section 3.2 of this paper.

To summarize, Table 3 shows the step-by-step emergence of the S-HRM model, documenting the authors' analytical work. They highlight the eight S-HRM practices and suggest facultative keywords and wider context for further quantitative-qualitative content analysis of non-financial reports.

The following Section 3.2. introduces the S-HRM Practices Model which is derived from the Analytical Framework and can be further used to analyze various types of non-financial reports when searching for optimum, efficient and harmonized S-HRM disclosure as described in the *Methodology*.

**Table 3.** Sustainable Human Resource Management: Analytical Framework.

| GRI Standards/S-HRM Practices (1–8) Defined | | SHRM Characteristics Compliance * Yes/No | Research Relevance Yes/No | Report Monitoring Other Idicators (Not Listed Elsewhere) |
|---|---|---|---|---|
| **S-HRM: 1 Employee focus and long-term employment strategy** | | Focus: Long-term strategy and growth | | |
| GRI Employment, General Disclosures, Management Approach | New employee hires and employee turnover, 401-1 | New hires, turnover (age, gender, region) | Yes/A * Long term-orientation | Yes | Monitoring employee churn, ratios of age, gender Recruitment Identification of future HR availability Monitoring labour market Reaction to demogr. changes |
| | Parental leave, 401-3 | Employees (entitled, took, returned after) parental leave, retention rates (by gender) | Yes/A * Long term-orientation | Yes | |
| | Information on employees and other workers, 102-8 | Permanent/ temporary contracts and full-time/ part-time jobs (by gender, region). Nature and scale of work of non-employees (outsourcing) Seasonal variations | Yes/A * Long term-orientation | Yes | |
| | The management approach and its components–strategy, 103-2 | Management approach to EMPLOYMENT and HR components: Policies, Goals and targets, Responsibilities, Resources. Processes, projects, programs and initiatives. | Yes/A * Long term-orientation | Yes | |
| **S-HRM: 2 Employee development and performance evaluation** | | Focus: Education, development and evaluation | | |
| GRI Training and education | Average hours of training per year per employee 404-1 | Average hours of training (gender, employee category) | Yes/F * Employee development | Yes | Type of education and training Increasing qualification Apprentices, graduates Internships Engagement Job rotation Succession planning Internal talent pool HR development Education structure of employees Identification of future needs (e.g. IT)-innovation |
| | Upgrading employee skills and transition assistance programs 404-2 | Type and scope of programs to upgrade employee skills. Transition assistance programmes-continued employability and the management of career endings (retirement, contract termination) | Yes/A * Long-term orientation Yes/F * Employee develop. Yes/G * External partnership | Yes | |
| | % of employees receiving regular perform. and career development reviews 404-3 | Received performance reviews and career development reviews by gender/ employee category | Yes/F * Employee develop. | Yes | |

**Table 3.** *Cont.*

| | GRI Standards/S-HRM Practices (1–8) Defined | | SHRM Characteristics Compliance * Yes/No | Research Relevance Yes/No | Report Monitoring Other Idicators (Not Listed Elsewhere) |
|---|---|---|---|---|---|
| | **S-HRM: 3 Labor-management relations and business ethics** | | Focus: Business ethics, social dialogue and cooperation | | |
| GRI Management Approach, Labor/Management Relations, Freedom of Association and Collective Bargaining | The management approach and its components–Strategy, 103–2 | Mgmt. approach to CORPORATE CULTURE AND ETHICS, L/M relations and employee empowerment Policies, Goals and targets, Responsibilities, Resources. Processes, projects, programs and initiatives. | Yes/J * Employee cooperation Yes/D * Employee participation and social dialogue | Yes | Cooperation Teamwork Labor union–benefits Workplace relations Corporate Governance Corporate Values |
| | Labor/ Management Relations 402-1 | Giving notices prior to the implementation of significant operational changes, if specified in collective agreements | Yes/D * Employee participation and social dialogue Yes/I * Compliance beyong labour regulations | Yes | |
| | Collective bargaining agreements 102-41 | Percentage under collective bargaining agreements. | Yes/D * Employee participation and social dialogue Yes/I * Compliance beyong labour regulations | Yes | |
| | Operations at risk for freedom of association and collec. bargaining violation 407-1 | Operations and suppliers in which workers' rights to exercise freedom of association or collective bargaining may be at risk. Measures to support rights for freedom of association and collective bargaining | Yes/D * Employee participation and social dialogue Yes/I * Compliance beyong labour regulations | Yes | |
| | **S-HRM: 4 Well-being and benefits** | | Focus: Employee care, motivation and benfits | | |
| GRI Employment | Benefits provided to full-time employees 401-2 | Life insurance, health care, disability and invalidity coverage, parental leave, retirement provision, stock ownership | Yes/I * Compl. beyong labour regulations Yes/E * Shared profitability Yes/B * Care of employees Yes/H * Flexibility | Yes | Meal vouchers, Phone, tablet/laptop, Points to Exchange (vacation, massage, sports, leisure) Well-being, worklife balance, flexitime, shared jobs, (flexible) work arrangements |

**Table 3.** *Cont.*

| | GRI Standards/S-HRM Practices (1–8) Defined | | SHRM Characteristics Compliance * Yes/No | Research Relevance Yes/No | Report Monitoring Other Idicators (Not Listed Elsewhere) |
|---|---|---|---|---|---|
| | **S-HRM: 5 Equality and non-discrimination** | | | Focus: Fair practices | |
| GRI Diversity and Equal Opportunity, Non-discrimination, Human Rights Assessment | Diversity of governance bodies and employees 405-1 | Individuals in governance bodies an employees per employee category (gender, age group: under 30/30–50/50 plus, minority) | Yes/K * Fairness and equality | Yes | Inclusion Employees with disability |
| | Ratio of basic salary and remuneration of women to men 405-2 | Salary and remuneration ratio of women/men ( by employee category/significant locations) | Yes/K * Fairness and equality | Yes | |
| | Discrimination and correct. actions taken 406-1 | Incidents of discrimination and actions taken | Yes/K * Fairness and equality | Yes | |
| | Human Rights Assessment 412-1–3 (410-1) | Operations subject to human rights reviews or human rights impact assessments Employee training on human rights policies or procedures (hours, percentage trained) Investment agreements including human rights clauses or that underwent human rights screening | Yes/K * Fairness and equality | Yes | |
| | Operations at risk for incidents of child labor 408-1 | Operations/suppliers at risk for child labor, young workers exposed to hazardous work. Measures taken. | Yes/K * Fairness and equality | No ** | |
| | Operations at risk for forced or compulsory labor 409-1 | Operations and suppliers at risk for incidents of forced or compulsory labor. Type of operation/ geographic areas. Measures taken. | Yes/K * Fairness and equality | No ** | |
| | **S-HRM: 6 Nurturing employee environmental sustainability** | | | Focus: Adopting in-house environmental sustainabilty | |
| GRI Management Approach | The management approach and its components-strategy 103-2 | Management approach to NURTURING EMPLOYEE ENVIRONMENTAL SUSTAINABILITY: Policies, Goals and targets, Responsibilities, Resources. Processes, projects, programs and initiatives. | Yes/C * Care of environment | Yes | Materials Waste Energy Water Environmental compliane Employee assessment |

**Table 3.** *Cont.*

| | GRI Standards/S-HRM Practices (1–8) Defined | | SHRM Characteristics Compliance * Yes/No | Research Relevance Yes/No | Report Monitoring Other Idicators (Not Listed Elsewhere) |
|---|---|---|---|---|---|
| | **S-HRM: 7 Cooperation with external stakeholders** | | Focus: Partnership, communities and responsibility | | |
| GRI Local communities Rights of indigenous peoples | Operations with local community, impact assessments, and development programs 413–1 | Operations with local community, impact assessments, and/or development programs, public disclosure of results of environmental and social impact assessments | Yes/A * Long term-orientation Yes/ G * External partnership | Yes | External partnerships–social aspects Cooperation with educational institutions (universities) Clusters CSR/ sustainability social aspect towards community Clusters |
| | Potential negative impacts on local communities 413-2 | Operations with significant actual and potential negative impacts on local communities | Yes/A * Long term-orientation Yes/ G * External partnership | Yes | |
| | Violations involving rights of indigenous peoples 411-1 | Number of identified incidents of violations involving the rights of indigenous peoples | Yes/A * Long term-orientation Yes/G * External partnership | No ** | |
| | **S-HRM: 8 Safety and health at work** | | Focus: Promotion, prevention & adoption of health and safety | | |
| GRI Occupational Health and Safety | Occupational health and safety 403-1–10 | Occupational health and safety management systems. Hazard identification, risk assessment, and incident investigation. Occupational health services. Worker participation, consultation, and communication on occupational health and safety. Worker training on occupational health and safety Promotion of worker health. Prevention and mitigation of occupational health and safety impacts directly linked by business relationships Workers covered by an occupational health and safety management system. Work-related injuries and ill health. | Yes/B * Care of employees | Yes | Work-related injuries Fatalities Access to non-occupational medical and healthcare services Health promotion programs |

* SHRM characteristics based on a thorough SHRM literature review and a SCOPUS study by (2018) * Stankeviciute and Savaneviciene. A—Long term-orientation, B—Care of employees, C—Care of environment, D—Employee participation, E—Profitability (shared), F—Employee development, G—External partnership, H—Flexibility, I—Compliance beyond labor regulations, J—Employee cooperation, K—Fairness and equality ** topics to be excluded from the research, mostly N/A to Czech companies or marginally applicable.

*3.2. Defining Corporate S-HRM Practices Model*

Based on the GRI standards, S-HRM literature review and the S-HRM Analytical Framework presented in the previous sub-chapter, eight key areas of sustainable human resource management practices emerged from the analysis. These are presented as follows with no preference of order. They were shortened from the S-HRM Analytical Framework for the purposes of the S-HRM Practices Model (Figure 3):

- Employee development and evaluation;
- Health and safety;
- External stakeholders and partners;
- Focus on employees and long-term strategy;
- Employee environmental sustainability;
- Ethics and labor management relations;
- Well-being and benefits;
- Non-discrimination and equality.

The category of human rights and dignity at work might be considered and added to the proposed S-HRM Model in harmony with the local conditions especially for multi-national corporations and companies with an international supply chain where human rights may be at risk (child labor, indigenous peoples' rights at risk, or the dangers of sweatshop labor).

In the Czech Republic, while designing the harmonized approach to the analysis of S-HRM reporting, the category of human rights and dignity at work was excluded from the model since this category is marginally applicable. In the Czech Republic, the respect for human rights is primarily ensured by the Labor Code.

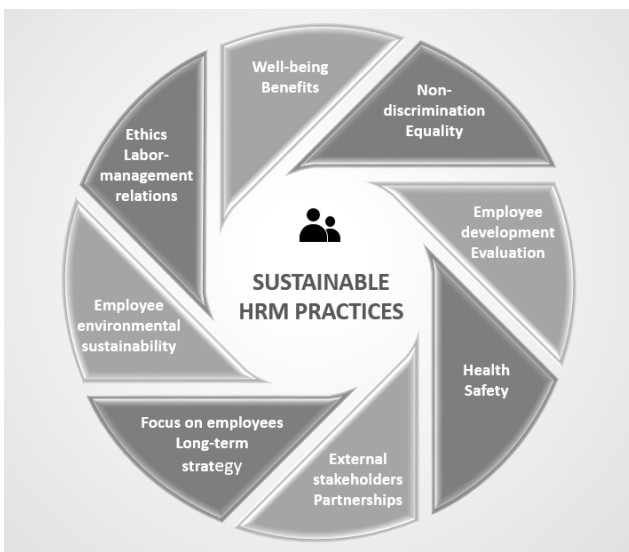

**Figure 3.** Sustainable HRM Practices Model.

The evaluation feedback of the Sustainable HRM practices, whose aggregate results are presented in Table 4, was elicited and collected in January–February 2021 from the panel of experts. The panelists were selected based on the predefined criteria with the requirement to fulfill at least four of the given six criteria, which were as follows:

1. *Minimum of 10 years of active participation and experience in the field;*
2. *University education (master/doctoral or higher);*
3. *Academia—a renowned expert in the field (specified below);*
4. *Active practitioner—expert in the field (specified below);*
5. *Active in research and publishing;*
6. *Field of work * (min. one):*

* *HR management, sustainable HR management, work relationships, HR and leadership development;*
* *Corporate Social Responsibility, Sustainability, Corporate Governance;*
* *Communication corporate strategies, Reporting;*
* *Organizational development, Organizational settings*

The panel of eight subject matter experts, representing both academia and corporate and private business sectors, fulfilling at least four from the six predefined criteria, were chosen based on their expertise in theory and/or practice, their education, research, publishing achievements and active participation in the above mentioned fields.

The experts come from the following institutions: University of Minnesota (USA), London South Bank University (UK), University of Essex (UK), a freelance expert specializing in corporate training and HR (UK), University of West Bohemia (CR), Prague University of Economics and Business (CR), Masaryk University (CR), and University of Finance and Administration (CR).

The number of experts not only conforms to Lawshe's model but exceeds the minimum required number of panelists. Thus, the amount of people validating the model is very germane.

The experts had been asked to evaluate the criteria on the three-point scale and mark them as: essential, not essential or non-applicable in terms of S-HRM. Having received the answers from the panelists, the authors of the paper ran the content validity test and calculated the content validity score for each criterion. The calculations' results in Table 4 show that each criterion received a CVR values above the minimum score (can be compared to Table 2), which means that all the indicators—the defined key areas of S-HRM in companies—are valid.

Therefore, the S-HRM Practices Model is validated for further use and can serve as a basis and checklist of S-HRM corporate practices. In the next step, it can be extended by subcategories and keywords (based on the Analytical Framework) and serve as an evaluation tool for quantitative content analysis of corporate non-financial reports. A graphic representation of the S-HRM practices was presented as Figure 3.

**Table 4.** Content validity of the Sustainable HRM Practices Model.

| Sustainable HRM Practices | Essential | Not Essential | Non-Applicable | CVR Score | Note |
|---|---|---|---|---|---|
| Employee focus and long-term employment strategy | 8 | - | - | 0.99 | Valid |
| Employee development and evaluation | 8 | - | - | 0.99 | Valid |
| Labor-management relations and business ethics | 7 | 1 | - | 0.75 | Valid |
| Well-being and benefits | 7 | 1 | - | 0.75 | Valid |
| Equality and non-discrimination | 7 | 1 | - | 0.75 | Valid |
| Nurturing employee environmental sustainability | 7 | 1 | - | 0.75 | Valid |
| Cooperation with external stakeholders | 7 | 1 | - | 0.75 | Valid |
| Safety and health at work | 8 | - | - | 0.99 | Valid |
| CONTENT VALIDITY INDEX: | - | - | - | 0.84 | Valid |

To elaborate the topic in detail, in Sections 3.2.1–3.2.8, the individual practices are further described and supported by the literature.

### 3.2.1. Employee Focus and Long-Term Strategy

Companies and other business entities taking care of their key assets, i.e., their employees, exercise in practice the idea of sustainable entrepreneurship. It is crucial that organizations focus their attention primarily on a long-term employee strategy including monitoring labor markets, demographic changes, revising recruitment practices as well as on the employee agenda concerning employee turnover, age and gender ratio, new hires, talent management, retention rates, parental leave, levels of outsourcing, etc. Sivathanu and Pillai (2018) also look into the near future and point out that organizations will need successful strategies to face transformational challenges of the Industry 4.0, which may perhaps result in more efficient and leaner HR teams.

Wharton management professor Michael Useem, a co-author of the book *Go Long: Why Long-Term Thinking Is Your Best Short-Term Strategy* (Carey et al. 2018), reminds the readers about the situation at the turn of 21st century when a number of prosperous companies "tried to reinvent the world a bit by focusing more on long-term goals and less on next quarter's earning", naming Apple, Amazon, Facebook, Google and Microsoft, i.e., companies that overcame their competitors. However, leadership strategy and tactics resisting short-term goals and pressures need to be reinvented through a human focus (Human Capital Trends Report 2019). The success of these companies could not have been achieved without a long-term seamless integration of an S-HRM strategy into the overall business strategy.

### 3.2.2. Employee Development and Evaluation

Employee development plays one of the fundamental roles of sustainable HR management. This task is rather complex in view of new challenges that are ever emerging, predominantly in connection with technological advancement. Often, a competitive advantage of the company lies within shared knowledge, employee skills and their abilities to grow, innovate or accept technological changes. Špacek and Vacík (2016, p. 65) claim that "a company possesses key competencies as fundamental factors to generate competitive advantage. These competences are unique and difficult to imitate. One of the most significant is the ability to innovate".

Business leaders are fully aware of the fact that timely employee upskilling for near future business environment is a core of success., e.g., getting ready for AI-enabled environment is a question of a completely new type of training (Sage-Gavin et al. 2019). Thus, apart from standard employee evaluation systems, future evaluation of employee performance will have to include criteria including, e.g., positive attitude of employees towards technologically oriented types of training and willingness and readiness "to go an extra mile". With regard to complex technological challenges, companies and their employees will always have to find common ground in reskilling and upskilling training programs' participation (Knihová and Hronová 2019).

### 3.2.3. Labor Management Relations and Business Ethics

Labor management relations and the operational climate perceived by employees has been of growing importance for all parties involved. Be it labor unions, labor-management negotiations or employee involvement programs, business ethics plays an important role (Jha and Singh 2019). In addition, industrial relations and business sector climate are an important part of GRI standards. The question of business ethics and ethical leadership has been discussed in a recent article published by Harvard Business Review. In this article, Max H. Bazerman, a professor of Business Administration at Harvard Business School, highlights that "People follow the behavior of others, particularly those in positions of power and prestige. Employees in organizations with ethical leaders can be expected to behave more ethically themselves", adding that it is the business leaders who shape the business environment, or decide on the environmental footprint of their business operations. The author concludes his article by an appeal: "Together we can do our best to be better." (Bazerman 2020).

### 3.2.4. Well-Being and Benefits

The GRI standard of well-being and benefits concerns benefits provided to full-time employees (401–2). They represent "a full care bundle" of perks from meal vouchers to tablets/smartphones and/or extended vacations. All these benefits boost loyalty and motivate employees. Pressure on high performance in organizations can cause mental problems. Thus, maintaining both mental and physical well-being is crucial in the workplace. Working climate shall definitely support all age levels and create satisfying and intrinsically valuable environment for employees (van Dam et al. 2017).

However, these turbulent times bring some inevitable changes. These are related to flexible working arrangements, remote working, home office and hybrid work arrangements, the category of well-beings and employee benefits. When asked about flexible working, more than 15,000 business people across over 100 nations confirmed that 50% of employees globally were working outside their main office for at least 2.5 days a week (International Workplace Group 2019). The dynamics of changes in work arrangements is highly probable to cause inevitable alterations to the reading of the GRI standard 401–2. The main reason is that the growing proportion of the workforce which is already working or is expected to work in flexible work arrangements can neither be excluded from the reporting, nor discriminated against full-time employees. We cannot expect that the world of work will be the same as in the pre-pandemic time.

### 3.2.5. Equality and Non-Discrimination

The GRI standards discussed here include a number of topics pertaining to various issues of equality and non-discrimination. Among the most frequented ones there are gender equality (Mehnert 2019), racial injustice, salary and remuneration differences between men and women, minorities, child labor and human rights. Geographical location of a particular business entity (not only its headquarters but above all its plants' location) will influence the selection of individual parameters for the reporting., e.g., while child labor does not exist in the Czech Republic, it may be an issue in some parts of the world. The highest levels of flexibility, a cognitive-emotional approach and, above all, empathy are recommended here regarding whether or not to include a certain GRI standard into the report.

### 3.2.6. Nurturing Employee Environmental Sustainability

The GRI standard 409–1 deals with policies adopting in-house environmental sustainability in relation to employees, their duties and responsibilities. Be it a procurement manager or R&D department, the goal for everybody is the same: all decisions should be harmless to environment and the environmentally friendly solution is always preferred. Sometimes, such a decision represents higher costs, but it should not be a barrier. Corporate sustainability is always high on the agenda of business managers with decision-making power. Well-designed S-HRM practices may help the adoption of green practices (Renwick et al. 2012; Wagner 2012). Additionally, policies in the fields of training and development, recruitment, relations in the workplace, pay and reward and others are considered powerful tools to align employees with green organizational strategy (Arulrajah et al.). Many researchers examine various aspects of HRM practices focusing on the 'fragility of ecosystems' and the negative effects of business activities highlighting the evolution of S-HRM along with its implications for employees (Randev and Jha 2019). Chaudhary (2018) in his study found that green HRM of a company was related significantly to the job pursuit intention of prospective applicants. Well-designed S-HRM practices may help the adoption of green practices (Renwick et al. 2012; Wagner 2012).

### 3.2.7. Cooperation with External Stakeholders

Cooperation and efficient communication with stakeholders are vital. The highest levels of corporate diplomacy are up to standard as external publics are diverse. "The set of potential stakeholders can seem overwhelming. Screening based on influence is thus required. Include in your database any stakeholder who can ruin your day. That potential is a function of how much power stakeholder have and how much your project matters to them" (Henisz 2017, p. 22). The GRI 413–1 deals with the social aspects of external partnerships, CSR towards local communities, clusters and/or cooperation with educational institutions, etc. There are many items to include into non-financial reporting. Being active in these areas and reporting on them has the capacity to positively influence a company's image and boost good reputation as well as the overall perception of a brand or organization.

### 3.2.8. Safety and Health at Work

The International Health Organization, aiming to create worldwide awareness of the dimensions and consequences of work-related accidents, injuries and diseases, publishes appalling numbers on its website: "Every day, people die as a result of occupational accidents or work-related diseases—more than 2.78 million deaths per year" (Safety and Health at Work 2021). Safety and health represent an important aspect of GRI reporting. Promotion, prevention and adoption of health and safety precautions may vary in different countries, but mitigation of occupational and safety impacts represents the same value elsewhere. Employee training focused on occupational health and safety is obligatory in many countries. Employers fund various health programs; however, barrier-free access to health programs provided by employers should not be considered as an employee benefit.

### 4. Discussion

In the discussion section of this paper, the authors would like to focus on the interpretation of the results and main research findings in relation to the research question.

The goal of this paper was to propose the eight-item S-HRM Practices Model serving as an instrument for harmonized reporting and the needs of academia and practitioners alike. The Model has been devised based on the GRI standards' comparative analysis providing an in-depth qualitative insight, literature review, interpretation of studies on S-HRM and Lawshe's content validity approach. In addition, the study shall contribute to the scientific literature, bridging the existing gap between the theoretical studies purely on S-HRM and more practically oriented studies on sustainable reporting.

The authors' main motivation to work on this study was the palpable discrepancy between the voluntary character of sustainability reporting (except for the EU Directive 204/95/EU) and the logical needs of business practice calling for harmonized tools. Performing a meticulous literature review helped the authors to precisely identify the existing research gap, formulate the research question, and propose a solution valuable both for academia and business practice.

The main findings analyzed and discussed below represent answers to the research question (RQ): *What are key areas for auditing sustainable HR practices in non-financial statements?* The portfolio of selected research tools enabled to fulfill the purpose, reach the goal and arrive at the desired answers to the RQ, thus revealing the main findings.

Shortly restating the results and the main findings of this study (Figure 4), the S-HRM Analytical Framework was elaborated, at first mainly based on GRI 3, 4 and GRI Standards, a study called *HRM's Role in Corporate Social and Environmental* Sustainability (2012) and inspired by Stankevičiūtė and Savanevičienė's (2018) theoretical paper.

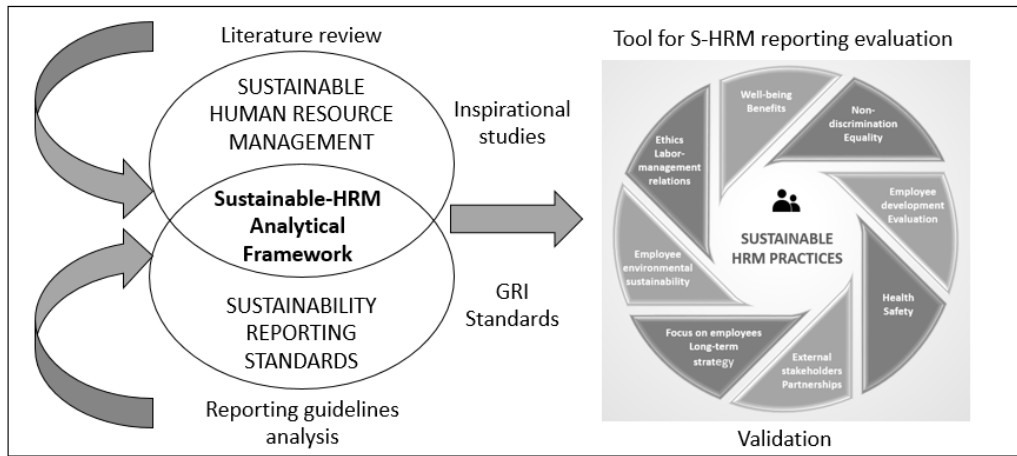

**Figure 4.** Methodological model findings.

From the study on HRM's role in sustainability, the so-called indicator GRI categories with direct relevance to HRM (originally in the study as follows: economic, environment, labor practices, human rights, society and product responsibility) were carefully examined with the attention paid to their affiliation to S-HRM practices in accordance with the aims of this study and the Stankeviciute and Savanevicience's theoretical findings based on the S-HRM literature review.

According to these authors (Stankevičiūtė and Savanevičienė 2018), the characteristics of S-HRM which emerged from theoretical studies are as follows: long term-orientation; care of employees; care of environment; employee participation; profitability (shared); employee development; external partnership; flexibility; compliance beyond labor regulations; employee cooperation; and fairness and equality. All of them have been found to be in compliance with the aims of this paper (marked as A–K in Table 3) and checked and grouped most appropriately according to the GRI standards. Having prepared the S-HRM Analytical Framework, the authors elaborated and devised the S-HRM Practices Model presented in Section 3.2 of this study. Consequently, the second Scopus indexed paper by Jang and Ardichvili (2020) was used as a benchmark tool for comparing the S-HRM 1–8 categories defined in this study (and validated by the panel of experts) with the CSR and sustainability-related themes most frequently occurring in the corporate reports/scientific papers analyzed by these authors. Their categories focused on HR development—close to but not the very exact aim as this paper's. Their defined categories specified on the basis of reports on HR development were: Diversity, equity and inclusion; community engagement; work–life balance; employee long-term growth and development; performance management; business ethics and ethical culture; and raising CSR awareness.

Interestingly, both studies do not implicitly define the human rights theme. Based on the reasoning about the purpose and aim of their study, the occupational health and safety category was excluded, as it lacked explanatory power by Jang and Ardichvili (2020). Their seven defined and above mentioned categories based on content analysis of the corporate sustainability/CSR reports used slightly different wording and mildly different distribution, but they are corresponding in most of their content with the results and main findings of this study. All three papers were used as a basis and inspiration for this original study, as none of them could be fully replicated. All of them were components which helped to create the added value of this paper—the development of the analytical framework and the model presented in the *Results* section of this study.

With the use of the generated S-HRM Analytical Framework mentioned above, the answer to the research question was presented, offering the main findings, namely the eight main categories of S-HRM practices which were derived and presented in the form of the S-HRM Practices Model (Figure 3), as follows: employee focus and long-term strategy, employee development and evaluation, labor management relations and business ethics, well-being and benefits, equality and non-discrimination, nurturing employee environmental sustainability, cooperation with external stakeholders and safety and health at work (Table 3). The categories of S-HRM practices were validated with the use of the Lawshe's (1975) content validity test based on evaluations of an international panel of experts publishing on HR, CSR/sustainability reporting, corporate governance and sustainable HRM.

Besides the validated model, the purpose of this paper was to extend and update the literature on sustainable human resource management and state-of-the-art reporting guidelines. Its aim was to close the gap in the specific research avenue between the theoretical studies purely on S-HRM and more practically oriented studies on sustainable reporting.

The authors are aware that the research may have several limitations. The first is partially a geographical limitation given by the focus on the Czech Republic in terms of the reporting practices. The second is the narrowing of the study with the exclusion the human rights category, as in the CR this is mostly covered by the Labor Code. Finally, the study does not address the specific economic benefits of sustainable corporate be-

havior with a focus on S-HRM and does not engage in the purely economic field of HR sustainability research.

One category which could be considered and added to the S-HRM Model for multinational corporations and companies with an international supply chain based on local conditions where human rights may be at risk (e.g., because of the child labor, indigenous peoples' rights at risk, or the dangers of sweatshop labor) is human rights and dignity at work. This was excluded from the list as in the Czech Republic, where the tool is planned to be used further for the analysis of S-HRM reporting, this is marginally applicable, as the respect for human rights is primarily ensured by the Labor Code.

Located in the heart of Europe, the Czech Republic is a preferred geographical location of many global brands. Many subsidiaries of global companies have their operations in the Czech Republic, including Coca-Cola or Harley Davidson, to name a few. These branches operate locally here, and very often, their managers tailor their business decisions to the local environment, applying the idea of glocalization into their business practices. Thus, the standardized and validated Sustainable HRM Practices Model to be used in the evaluation process of sustainability reports with a focus on S-HRM has been proposed by the authors of this study based on the prevailing situation in Central Europe. It encompasses the typical model situations both in global and local companies operating in the Czech Republic, which is not much different from the neighboring countries. Therefore, the model does not include the issues related to human rights and their possible violation because this concern is not typically seen in the HRM practices in local and Central European business environments, as mentioned above.

However, violation of human rights in employment could be an issue at the global level. The authors of this study are convinced that the devised model, due to the streamlined selection of eight areas of S-HRM practices only, may serve as a self-explanatory guide for many companies and business entities operating in similar conditions. The model will help them in their efforts to introduce a comprehensible and easy to use mode of sustainable HRM practices and join the community of companies that pro-actively promote sustainable non-financial reporting and believe in its social impact.

Implications for theory and academia are obvious. By publishing research findings, the knowledge base among academia on the S-HRM, its practices and reporting is widening and the identified gap is narrowing. Further development of the model is possible as well as replication of its use for S-HRM analysis in research. Building on ideas is welcome and beneficial. In tertiary pedagogy, teaching should have a practical overlap. The model can be used as a basis for creating learning materials, for example, for the application of critical thinking, creativity, and argumentation skills in creating a communication strategy of the company with the inclusion of S-HRM practices. This way, university students can gain the skills that current employers are calling for. Managers-to-be may learn about the concept firsthand while studying at university.

Implications for the overlapping areas of theory and practice. If the academic sphere writes studies on the topic, it may ease the transfer of this complex issue into practice. Academia can adopt a proactive attitude and help create preparatory steps with KPMG and other organizations that strive to create standardized methodologies. This can be done in parallel, ideally on the basis of collaboration of academia and practice.

Implications for practice. The proposed model enables harmonizing the tools for S-HRM practices evaluation in sustainability reports. For instance, in the Czech Republic, when incorporating the directive of the EU on reporting, requirements were not tightened and the obligation to report was not extended beyond the circle of public interest organizations. Companies were left with space for voluntary reporting. Growing demand for non-financial indicators from customers, investors and the public is visible, yet a harmonized tool is missing, even though GRI Standards prevail as the most widely used reporting guidelines (KPMG 2020). Other implications and advantages of the main findings are as follows. With the use of the model, a responsible employer brand can be identified with greater ease and various stakeholder groups may benefit from it. The tool

brings added value as it can influence behavior of managers who would be willing to incorporate a tailored S-HRM strategy considering employees a valuable asset into their strategic documents. It may impact the competitive environment, bearing in mind that it is hard to replicate a well-prepared and in-detail elaborated S-HRM strategy. Last but not least, the information revealed based on the S-HRM Practices Model will most likely impact investors who sift through potential investment opportunities and are increasingly demanding the non-financial indicators to serve their decision-making process.

The authors wished to broaden and update the literature on S-HRM and its reporting. They recommend further research on reporting the S-HRM practices. More extensive use of the S-HRM Analytical Framework and the Sustainable HRM Practices Model for qualitative analyses of corporate documents can be proposed as well as extension of the research into other areas—e.g., research on the causality of sustainable S-HRM practices and employee engagement.

## 5. Conclusions

We are living in turbulent times. Technological advancement is progressing at a very high speed and innovation brings ever changing conditions and influences processes in companies and organization on all levels. The external legal, political and socioeconomic forces of the macro environment cause uncertainties. Pressure on economic outcomes and profitability of a company as a whole as well as on individual performance of their employees can cause human resources strive.

Fortunately, based on a gradual development, companies and organizations are increasingly focusing on the incorporation of sustainable activities into their daily operations as well as business strategies and business plans. HR plays a key role in this process. The concept of sustainability has penetrated all management functions and has gained attention from practitioners and academia alike. Proper HR functioning within the organization can help in utilizing the HR tools and help embedding the sustainability strategy into organization.

Hand in hand with the incorporation of sustainability into organization on all levels so that all internal stakeholders act responsibly and sustainably, the organizations seek to apply ethical and sustainable approaches towards their key assets, employees, too. Here lies potential for a growing and sustainable company with skilled, engaged and satisfied employees who are willing to learn, grow, stay loyal and sustain the dangers of the external turbulent environment together with the company. Many companies would not have achieved success without a long-term seamless integration of S-HRM practices into their overall business strategies.

In an attempt to address the question of the significance of this research, the authors claim that being cognizant of the introduced Sustainable HRM Practices Model and its application is the first important step on the path towards improving the relationship with stakeholders. Among them, employees, owners, investors and the local community are the most important. By getting acquainted with the annual S-HRM report, employees will appreciate the sustainable HR management that directly influences their lives and career advancement. An in-depth analysis of the S-HRM report of a company may persuade potential investors to channel their investments because they are persuaded that the company has unique management with acumen which may be a guarantee of future achievements and economic growth. Appreciating the already carried out activities of sustainable character, local communities may come with new ideas for cooperation. In return, it may boost the employer brand, brand image and excellent reputation of the company.

It is the opinion of the authors that this study should be seen in a broader context. Sustainable human resource management is a concept of much higher significance. The organizational dynamics is largely influenced by the workforce. A holistic perspective, cognitive-emotional approach and particular context-sensitivity should be high on the agenda of the HR management of companies. Sustainable HRM is a complex agenda. Therefore, close cooperation between academia and subject matter experts from practice is

much needed to enhance sustainable non-financial reporting as an efficient communication channel between the company and its stakeholders. The proposed model may serve as an ingenious instrument and contribution to sustainable management theories as well as sustainable management practices.

**Author Contributions:** Conceptualization, Š.H. and M.Š.; methodology, Š.H. and M.Š.; validation, Š.H. and M.Š.; formal analysis, Š.H. and M.Š.; resources, Š.H. and M.Š.; data curation, Š.H. and M.Š.; writing—original draft preparation, Š.H. and M.Š.; writing—review and editing, Š.H. and M.Š. All authors have read and agreed to the published version of the manuscript.

**Funding:** This research received no external funding.

**Institutional Review Board Statement:** Not applicable.

**Informed Consent Statement:** Not applicable.

**Data Availability Statement:** Publicly available data was used. All sources are cited in the References.

**Conflicts of Interest:** The authors declare no conflict of interest.

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
