# Peer review of "Sustainable HRM Practices in Corporate Reporting"

_economies, doi:10.3390/economies9020075_

Round 1

Reviewer 1 Report

The paper titled “Sustainable HRM Practices in Corporate Reporting” is interesting and well written, and I think it has potential to make a valuable contribution to the field Human Resources. I would like to highlight that it clearly states its objectives and connect them with the existing gap that it intends to cover. Additionally, the manuscript really emphasizes the important implications that it has. However, I believe there are some areas where improvements must be made. I detail specific concerns and comments below.

First, although I really like the chronology that is showed in the introduction section (as it helps to understand how this area has been developed during the last decades), I also consider that some paragraphs read as excessively descriptive (I have the feeling that there is lack of integration of the different ideas). In other words, I see some paragraphs (e.g., lines 62-84) that offer more a list of events than a story. I would improve this section by highlighting the most important events in the development of S-HRM and connecting better the different events.

Second, in page 2 lines 98-99 you mention “With the use of the generated S-HRM analytical model mentioned above, 8 main S-HRM categories of S-HRM practices have been derived”; however, I do not see the rationale behind this 8 categories (where they come from? Any reference may support this classification?) I agree you validate these categories, but I would like to understand why you propose these categories and not others.

Third, it just doesn’t convince me the general structure of the article. I do not understand why sections 3 and 4 appear after section 2 (methodology). In my opinion sections 3 and 4 should be part of the introduction section or, at least, appear before the methodology section.

Finally, I consider that the manuscript should be reviewed in order to correct some minor mistakes, such as:

  • Line 55: “human resources” must be “HR”
  • Line 58: “as the adoption” must be “As the adoption”
  • Line 63: “on S-HRM e.g. in” must be “on S-HRM - e.g. in”
  • Table 2 format (it appears cut into too many sections)

Reviewer 2 Report

Dear authors, you have chosen an interesting and actual research area. I believe that the importance of this topic will increase  among academics and practitioners. I believe that your paper has potential but it needs a lot more work to get it published. My main concerns are as follows:

-the paper lacks clear structure  which is typically in academic articles. Please read good academic papers and follow the main structure (the introduction, the literature review, methodology, results etc.)

-The introduction includes relevant literature, but you should do the literature review  elsewhere and be more focused in introduction. At current form, you mainly list of publications.

--What is research question- why it is important in theory and practice?

--What do we know, what do we don't know?

--What we will learn as researchers?

Why the topic is interesting (I suggest that you narrow your scope more to s-hrm). Actually, you seem to have read the main literature of the topic. I would encourage you to read also Järlström, Saru & Vanhala (2008): Sustainable human resource management with salience of stakeholders published in the Journal of Business Ethics (152, pp. 703-724). Your results seem to have something in common with their framework of sustainable HRM and with those practices. 

-I would advice you to have a clear focus first on S-HRM concept and then jump to reporting practices. 

-is your analysis deductive or inductive or something between these (abductive)? Somehow, I couldn't see a clear link between your analysis and theoretical model. Your contribution needs to be more clear.

Reviewer 3 Report

Dear Authors.

This article aspires to "broaden and update the literature on S-HRM and its reporting" - lines 15-16. Yet, there is not novelty in it - I can not see any value added but citing others' opinions and approaches.

The methodology section - what were the criteria of "expert" selection and what constitutes being an expert? Moreover, why we should rely on their opinions? Why 8 "experts" not 80? 

The similar issue is related to literature selection. I strongly recommend SLR or bibliometrics.

f.e.

Boell, S.; Cecez-Kecmanovic, D. On being ‘systematic’ in literature reviews in IS. J. Inf. Technol. 2015, 30, 161–173, doi:10.1057/jit.2014.26.

Watson, R. Beyond being systematic in literature reviews in IS. J. Inf. Technol. 2015, 30, 185–187, doi:10.1057/jit.2015.12.

Denyer, D.; Tranfield, D. Producting a systematic review. In The Sage Handbook of Organizational Research Methods; SAGE Publications Ltd.: London, UK, 2009; pp. 671–689.

Pritchard, A. Statistical Bibliography or Bibliometrics? J. Doc. 1969, 25, 348–349.

Herrera-Franco, G.; Montalvan-Burbano, N.; Carrion-Mero, P.; Apolo-Masache, B.; Jaya-Montalvo, M. Research Trends in Geoturism: A Bibliometric Analysis Using the Scopus Database. Geosciences 2020, 10, 397, doi:10.3390/geosciences10100379.

Janowski, A. Philanthropy and the Contribution of Andrew Carnegie to Corporate Social Responsibility, Sustainability 2021, 13, 155. https://doi.org/10.3390/ su13010155

Why Czech Republic should be the foundation for induction? What does this country representative?

Discussion and conclusions section.

The discussion in not usually perceived as a set of postulates....lines 559-579.

In this matter, it is extremely difficult to refer to Your conclusions and the review is pointless.

Please be so kind and reconsider

Round 2

Reviewer 2 Report

Thanks for your revised paper! I see some progress in paper development which is a positive issue. 

In abstract, line 19 - why entrepreneurship? Likewise, line 25-I am not convinced about this methodological approach (and/or as it is written here).

At the end of abstract, you mention that  finding was "the 8-item s-hrm practices model etc" compared to discussion(or conclusions) you say it was your goal/objective/purpose. 

I suppose that your literature review does not fit very well to the title. Your literature review is much broader and not very focused. As a reader, it is very difficult to follow. The so-called "red line" is missing. You could narrow it to make it more ready friendly.

Concerning literature review, I am not sure why it is so broad. You could have done literature review of sustainable HRM to be more focused. Now you list so many theoretical points, that it is unclear which is the theory you aim to contribute and how. One option might be, that you mention that topic is related to several theoretical viewpoints, but you selected the one which best support your empirical part (which seems to be related human resources and HR practices). 

I suppose that stakeholder theory might fit relatively well for your study, because the aim of sustainability reporting is mainly targeted for stakeholders. Could be raised up in introduction. As a reporting practices, how much you believe is so-called "window-dressing" or green-washing? You need to discuss of potential misleading of companies' reporting systems. How much we can trust on them and what kind of story can we read? are they telling positive issues and neglecting the negative ones?

Now, you are having less "type of listing" research, but I still see for some extent of it (e.g. lines 318-327). Try to ellaborate more if you import some research in your paper.

line.295 SD...?have you opened it earlier?

line 445...parenthesis?

chapter 1.2.5 -should there be HR reporting instead of sustainability reporting?

line 686 - why research question is here?

I suppose that literature review needs to be more focused before reading thoroughly the empirical part. There seems to be nice pictures and tables.

I suppose the research gate discussion is not relevant for academic paper. 

I hope that you re-write a paper to have it more reader friendly.

Good luck!

Reviewer 3 Report

Im not convinced. Yet, consider my approval as a credit for future better work.

Author Response

Dear Reviewer,

Thank you very much for giving your approval. It is highly appreciated.

Best regards,

The Authors.